# A Study of the Community Relationships Between Methanotrophs and Their Satellites Using Constraint-Based Modeling Approach

**DOI:** 10.3390/ijms252212469

**Published:** 2024-11-20

**Authors:** Maryam A. Esembaeva, Mikhail A. Kulyashov, Fedor A. Kolpakov, Ilya R. Akberdin

**Affiliations:** Department of Computational Biology, Scientific Center of Genetics and Life Sciences, Sirius University of Science and Technology, Sirius 354340, Russia; maryashaesembaeva@gmail.com (M.A.E.); kulyashov.ma@talantiuspeh.ru (M.A.K.); kolpakov.fa@talantiuspeh.ru (F.A.K.)

**Keywords:** community modeling, constraint-based modeling, methanotrophy, methane-utilizing bacteria, bacterial satellites, *Methylococcus capsulatus* (Bath), *Escherichia coli* W3110

## Abstract

Biotechnology continues to drive innovation in the production of pharmaceuticals, biofuels, and other valuable compounds, leveraging the power of microbial systems for enhanced yield and sustainability. Genome-scale metabolic (GSM) modeling has become an essential approach in this field, which enables a guide for targeting genetic modifications and the optimization of metabolic pathways for various industrial applications. While single-species GSM models have traditionally been employed to optimize strains like *Escherichia coli* and *Lactococcus lactis*, the integration of these models into community-based approaches is gaining momentum. Herein, we present a pipeline for community metabolic modeling with a user-friendly GUI, applying it to analyze interactions between *Methylococcus capsulatus*, a biotechnologically important methanotroph, and *Escherichia coli* W3110 under oxygen- and nitrogen-limited conditions. We constructed models with unmodified and homoserine-producing *E. coli* strains using the pipeline implemented in the original BioUML platform. The *E. coli* strain primarily utilized acetate from *M. capsulatus* under oxygen limitation. However, homoserine produced by *E. coli* significantly reduced acetate secretion and the community growth rate. This homoserine was taken up by *M. capsulatus*, converted to threonine, and further exchanged as amino acids. In nitrogen-limited modeling conditions, nitrate and ammonium exchanges supported the nitrogen needs, while carbon metabolism shifted to fumarate and malate, enhancing *E. coli* TCA cycle activity in both cases, with and without modifications. The presence of homoserine altered cross-feeding dynamics, boosting amino acid exchanges and increasing pyruvate availability for *M. capsulatus*. These findings suggest that homoserine production by *E. coli* optimizes resource use and has potential for enhancing microbial consortia productivity.

## 1. Introduction

Methanotrophic bacteria such as *Methylococcus capsulatus* [1] and *Methylotuvimicrobioum alcaliphilum* 20Z^R^ [2] have an active application in advanced biotechnology [3,4,5]. This group plays a critical role in the global effort to mitigate methane, a potent greenhouse gas. These microorganisms utilize methane as their primary carbon and energy source, which has significant implications for environmental management and biotechnological applications, including bioremediation, biofuel production, and the synthesis of high-value products, such as succinate and ectoine [2,5,6,7]. One of the widely used methanotrophs is *Methylococcus capsulatus*, which is actively harnessed in biotechnology for the production of single-cell protein [3,8,9]. However, at the same time, oxygen- and nitrogen-limited conditions often arise during methanotroph cultivation in bioreactors because these bacteria have high oxygen demands for methane oxidation and rapidly consume nitrogen for protein biosynthesis and growth [1,10]. The limited solubility of gases in liquid media restricts oxygen availability, while high cellular growth promptly depletes nitrogen sources. As a result, this negative impact from these conditions decreases the methanotroph’s growth by restricting methane oxidation and essential protein synthesis, resulting in reduced biomass production [2,11]. Operational management of these nutrients is crucial for sustaining high growth rates in bioreactors. Additionally, one of the by-products under these conditions is acetate, which is produced by methanotrophs and inhibits their growth because it disrupts efficient energy production, diverting resources from methane oxidation and limiting biomass synthesis [12,13,14]. This accumulation of acetate leads to metabolic inefficiencies that reduce the overall growth rates [1,15,16]. At the same time, it has been shown that the bacterium growth benefits from being part of a microbial consortium in bioreactor conditions, where interactions with other species enhance methane oxidation efficiency, provide additional metabolic pathways, and improve nutrient availability [1,2,16,17]. Co-cultivation can also mitigate the accumulation of toxic by-products, such as acetate or formate, which can inhibit methanotrophic activity if not removed by partner microorganisms.

Heterotrophic bacteria, such as species from the *Brevibacillus* and *Cupriavidus* genera, play a crucial role in consortia with *M. capsulatus*, significantly enhancing the growth, stability, and metabolic efficiency of these communities. These acetate-reducing bacteria engage in synergistic interactions with *M. capsulatus*, contributing to effective nutrient cycling, detoxification of harmful by-products, and improved production of valuable compounds [1]. Recent research has explored the potential of engineered *Escherichia coli* strains as satellite organisms in synthetic consortia with *Methylococcus capsulatus*. *E. coli* can be genetically modified to consume by-products of methane oxidation, such as acetate, thereby reducing the metabolic burden imposed on *Methylococcus capsulatus* and enhancing overall methane conversion efficiency. Notably, a study by [15] demonstrated that *E. coli* could utilize acetate and other organic acids produced by methanotrophs that promote methanotroph growth under co-cultivation conditions. Additionally, *E. coli* has the capability to produce valuable metabolites, including L-homoserine from acetate, potentially affecting the amino acid metabolism of methanotrophs [18].

The diversity of industrial microbial chassis and the corresponding increase in high-throughput experimental data for them have led to the intensified application of the system biology approach, which offers the opportunity to aggregate different omics data and analyze them in the framework of a union modeling system. The widely used constraint-based modeling offers the opportunity to reconstruct metabolic models on the genome-scale level (GSM models), providing an analysis of the bacterial metabolism at steady state [19,20]. For example, the *i*JO1366 GSM model for *Escherichia coli* has been used to optimize the production of biofuels and chemicals by predicting and modifying metabolic pathways [21]. *Lactococcus lactis* models were applied to enhance lactic acid production in dairy fermentation [22]. Single-species GSM models remain valuable in silico tools, especially when integrated with more advanced systems for biology approaches that incorporate multi-species interactions and dynamic environmental factors. These integrated approaches can significantly improve the predictive power of GSM models, making them more relevant for biotechnological applications [23,24]. However, the co-existence and joint dynamic functioning of several microbes during the fermentation on an industrial scale significantly impact and constrain the application of single-species GSM models in order to describe and predict community behavior. To address this, a modeling of microbial communities has emerged as an essential theoretical basis for industrial biotechnology [25,26,27]. Constraint-based metabolic community (CBMc) models integrate the metabolic capabilities of multiple species to predict collective behavior and metabolic interdependencies [26,28]. There is a list of widely used tools, including cFBA [25], MICOM [29], SteadyCom [30], PyCoMo [31], and Kbase [32,33], but all of them, excluding Kbase, have no web-application or user-friendly interface [28].

Despite the increasing number of genome-scale metabolic models (GSMs) for methanotrophs [34,35], its satellites [36,37,38] and software for community modeling, applications of the CBMc modeling approach to methanotrophic communities remain sparse, despite the fact that it is actively harnessed for modeling other different biotechnological communities [37,39,40]. Two recent studies for methanotrophic communities by Islam et al., 2020 [41], and Badr, He, and Wang, 2024 [42], have investigated microbial community dynamics, but comprehensive CBMc models that describe methanotroph-satellite interactions in bioreactor conditions are still lacking.

Herein, we demonstrate a workflow with GUI using the BioUML platform [43] for CBMc model reconstruction to analyze the interactions between *Methylococcus capsulatus* (*i*McBath model) and an engineered *E. coli* (*i*EC1372_W3110 model) strain under oxygen- and nitrogen-limited conditions in bioreactors. Our objective is to evaluate the potential of *E. coli* to reduce methanotroph by-products, particularly acetate, and assess its impact on the growth and metabolism of *Methylococcus capsulatus* using a community metabolic modeling approach.

## 2. Results

### 2.1. Modifications of the iMcBath Model

The *Methylococcus capsulatus* strain possesses two isoforms of methane monooxygenase (MMO), being particulate (pMMO) and soluble (sMMO), with their activity modulated by the copper concentration in the media. sMMO is active under low copper concentrations, whereas pMMO operates under high copper conditions [44,45,46]. Both enzymes catalyze the critical step in methane assimilation, which includes methane oxidation to methanol, requiring an electron donor. Based on Lieven and co-authors’ results [3], an uphill electron transfer mode in which the pMMO isoform is active was chosen. Despite the modifications made, the model did not correctly describe the production of by-products under oxygen- and nitrate-limiting conditions. However, according to the experimental data [1,15], *M. capsulatus* produces an acetate as a by-product under these limiting conditions. The secreted acetate leads to a decrease in pure methanotrophic culture growth. It causes strain co-cultivation with other satellite bacteria, which consume the excreted acetate as a carbon source. To determine the discrepancy caused between the model simulation and the experimental data for the strain, we made a comparison of the metaboli map with an alternative model for *M. capsulatus* (*i*MC535) and a model for *M. alcaliphilum* 20Z^R^ and change the reversibility of reactions in the central metabolic pathways. The activity of H_4_MPT, RuMP, EDD, and EMP pathways was also modified (details: Section 4). In contrast, the original model exhibited a flux of 12.3 mmol·gDCW^−1^·h^−1^ through the H_4_MPT pathway, which was significantly higher than the 6.1 mmol·gDCW^−1^·h^−1^ flux through the RuMP pathway. To address this discrepancy, we adjusted the upper and lower bounds of the FAEi reaction (the initial step in the H_4_MPT pathway) relative to the FALDtpp reaction. The flux of formaldehyde from FALDtpp into H_4_MPT was reduced by a factor of 2.2, resulting in an 8.39 mmol·gDCW^−1^·h^−1^ flux through H_4_MPT, consistent with a total FALDtpp flux of 18.5 mmol·gDCW^−1^·h^−1^. The EDD pathway is known to be more active than the EMP pathway in *M. capsulatus*. However, in the original model, the EDD flux was zero. To resolve this, we incrementally redirected the flux from the AH6PI reaction to the first reaction in the EDD pathway (details: Section 4). The model began producing acetate under the uphill electron transfer mechanism once the EDD pathway flux reached 6 mmol·gDCW^−1^·h^−1^, aligning with previous findings [3]. Furthermore, by setting fixed constraints on the studied metabolic pathways and reducing nitrogen/oxygen availability, we observed an increase in acetate production (Figure 1).

### 2.2. Modification of the E. coli GSM Model iEC1372_W3110 for L-Homoserine Production

The recent study by [18] demonstrated the potential for homoserine production in *Escherichia coli* through flux balance analysis of the *i*EC1372_W3110 model. To validate and expand upon these findings, we identified a metabolic configuration with maximal homoserine output, characterized by elevated fluxes through L-aspartase and acetyl-CoA synthetase pathways. Under these optimized conditions, the model predicted a homoserine production rate of 10.22 mmol·gDCW^−1^·h^−1^ (Table 1).

To explore the genetic interventions necessary to enhance the homoserine synthesis while maintaining non-zero biomass production, the OptFlux tool [47] was applied. Our analysis revealed that viable in silico modified strains were attainable only when the biomass was reduced to 40% of its wild-type level. We prioritized a variant exhibiting an eightfold flux increase through the pyrophosphate transport reaction (PItex) for further experimentation (Appendix A). The increased flux and active L-aspartase and acetyl-CoA synthetase pathways resulted in homoserine production at a rate of 5.365 mmol·gDCW^−1^·h^−1^, with a corresponding growth rate of 0.369 h^−1^ (Table 1).

### 2.3. Reconstruction of the Community Model for M. capsulatus and E. coli

To build a workflow for community modeling, Jupyter Notebook [48,49] was employed. The computational platform includes Python-based tools, such as COBRApy [50], MewPy [51], and PyCoMo [31], to give an opportunity for the reconstruction of the community GSM models. Four community models comprising modified models for *M. capsulatus* and *E. coli* were constructed based on the built-in PyCoMo in the BioUML platform (Figure 2). These models describe the *i*McBath model under oxygen-limited and nitrate-limited conditions, where the acetate produced as a by-product is utilized by *E. coli* as a carbon source. It is worth noting that two variants of *E. coli* models were used, as follows: the unmodified *i*EC1372_W3110 model and the modified version for homoserine production. These alternatives were considered to investigate the potential influence of homoserine (details: Section 4).

To achieve the oxygen and nitrogen limitation conditions, we reduced their consumption rates in the model growth medium. To simulate oxygen limitation, its consumption rate was restricted to 22.152 mmol·gDCW^−1^ (based on an oxygen/methane ratio of 1.2). Additionally, to establish oxygen-limited conditions in the community, a strict boundary was set on methane consumption for *i*McBath (18.46 mmol·gDCW^−1^) to maintain the necessary oxygen/methane ratio. To limit the nitrogen content, the consumption rate was reduced to 1.838 mmol·gDCW^−1^ (60% of the total required for normal growth in the *i*McBath and *i*EC1372_W3110 models).

### 2.4. Analysis of the Interactions Between M. capsulatus and E. coli in the Microbial Community

Oxygen-limited conditions result in distinct growth rates for the unmodified *i*EC1372_W3110 (0.225 h^−1^) model and the homoserine-producing *i*EC1372_W3110 model (0.186 h^−1^; Table 2). In contrast, the community models with both the unmodified and modified *E. coli* exhibited identical growth rates of 0.217 h^−1^ under nitrate-limited conditions (Table 2). To identify differences in metabolites and reaction flux distribution in different versions of the community model, the cross-feeding metabolites and corresponding fluxes were visualized using ScyNet [52].

#### 2.4.1. Oxygen-Limited Conditions

To simulate oxygen-limited conditions, the methane transport was restricted in the *i*McBath model (details: Section 4), while the oxygen consumption rate from the medium was set to 22.152 mmol·gDCW^−1^·h^−1^. The flux balance analysis indicated that both modified and unmodified *E. coli* variants experienced microaerobic conditions, likely attributable to *M. capsulatus* dominating oxygen consumption. Additionally, nitrate reductase activity was observed in both community models, though notable differences in nitrogen sources utilized by each bacterium were found. In the model with the modified *E. coli*, both bacteria utilized NO_3_^−^ as their primary nitrogen source. *M. capsulatus* and *E. coli* consume NO_3_^−^ at rates of 8.467 and 2.879 mmol·gDCW^−1^·h^−1^, respectively, leading to NO_2_^−^ production in the medium by both organisms. Furthermore, *M. capsulatus* produced NH₄⁺ at a rate of 0.8473 mmol·gDCW^−1^·h^−1^, which was subsequently consumed by *E. coli* at a rate of 0.7755 mmol·gDCW^−1^·h^−1^. In contrast, in the community model with unmodified *E. coli*, nitrate reductase activity was only observed in the *M. capsulatus* model, where NO_3_^−^ was reduced to NO_2_^−^. This NO_2_^−^ was utilized by *E. coli* as a nitrogen source, using nitrite reductase (0.0589 mmol·gDCW^−1^·h^−1^). Despite the availability of NO_2_^−^, ammonium produced by *M. capsulatus* remained the primary nitrogen source for *E. coli* at a flux of 0.9811 mmol·gDCW^−1^·h^−1^ (Figure 3A,B, Appendix A). The increased production of ammonium by *M. capsulatus* could be contributing to the observed rise in the growth rate of the community (Appendix A).

The primary carbon source for *E. coli* in the community model incorporating the unmodified *i*EC1372_W3110 model is acetate produced by *M. capsulatus* (Appendix A), which is converted into acetyl-CoA and subsequently to pyruvate (Figure 3A). Meanwhile, methane remains the main carbon source for *M. capsulatus*. Notably, *E. coli* also produces glycerol (0.908 mmol·gDCW^−1^·h^−1^), which *M. capsulatus* metabolizes through the GLYCDx (glycerol dehydrogenase) reaction, resulting in the formation of dihydroxyacetone. This intermediate enters the DHAK (dihydroxyacetone kinase) reaction, producing dihydroxyacetone phosphate. This compound is processed in the TPI (triose-phosphate isomerase) reaction, generating glyceraldehyde 3-phosphate (Appendix A), which further converts in EMP pathway. *E. coli* in the model version also produces minor amounts of succinate, which serves as another carbon source within the community. In addition to classical carbon sources like sugars and C1 metabolites, amino acids can also serve as both carbon and nitrogen sources simultaneously. A range of cross-feeding amino acids between modeling strains in the analyzed community is observed. *M. capsulatus* produces threonine, while *E. coli* contributes aspartate, serine, and homoserine to the shared metabolic pool. Threonine produced by *M. capsulatus* is utilized by *E. coli* via the THRD_L (L-threonine deaminase) reaction, yielding 2-oxobutanoate and ammonia, whereas aspartate produced by *E. coli* is employed by *M. capsulatus* in the ASPTA (aspartate transaminase) reaction to synthesize L-glutamate and oxaloacetate, with the latter being an intermediate in the TCA cycle (Figure 3A). Serine from *E. coli* is used by *M. capsulatus* in the GHMT2r (glycine hydroxymethyltransferase) reaction for the production of methylenetetrahydrofolate, which is involved in purine metabolism. Interestingly, despite the absence of genetic modifications for L-homoserine production, the community model predicted that *E. coli* would produce homoserine at a flux of 0.282 mmol·gDCW^−1^·h^−1^. This homoserine is entirely absorbed by *M. capsulatus* and predominantly used in the HSK_GAPFILLING (homoserine kinase) reaction. The resulting O-phospho-L-homoserine is completely directed into the THRS (threonine synthase) reaction, leading to the production of phosphate and threonine, which are then consumed by *E. coli*, as described above (details: Appendix A). Additionally, the model predicts the exchange of water and phosphates between the community members, highlighting the intricate metabolic interdependencies within the community (a more detailed distribution of community model fluxes is provided in the pFBA results available on GitLab at the following link: https://gitlab.sirius-web.org/diploms/2023/esembaeva/-/blob/master/pFBA_data/o2_lim.txt?ref_type=heads (accessed on 18 November 2024).

Despite having identical environmental conditions, an analysis of the community model with the modified *i*EC1372_W3110 model revealed cross-feeding differences compared to the unmodified model. The carbon sources for *E. coli* in this community include acetate, malate, and glycerol, where malate serves as the primary source (1.144 mmol·gDCW^−1^·h^−1^), which *E. coli* utilized in the TCA cycle via the MDH (malate dehydrogenase) reaction (Figure 3B). It should be noted that acetate production decreased by 16.3 times in the modified community (from 3.343 to 0.205 mmol·gDCW^−1^·h^−1^) compared to the unmodified model version, which could explain the decreased growth rate of the community. Additionally, a small amount of glycerol (5.3 × 10^−^⁴ mmol·gDCW^−1^·h^−1^) produced by *M. capsulatus* is metabolized to dihydroxyacetone, which subsequently leads to the formation of fructose-6-phosphate, which feeds into the pentose phosphate pathway (details: Appendix A). Similar to the community with the unmodified *E. coli*, minor amounts of succinate are produced by *E. coli* and consumed by *M. capsulatus* (details: Appendix A).

As in the above-described model version, *E. coli* produces serine, which *M. capsulatus* utilizes in the GHMT2r (glycine hydroxymethyltransferase) reaction for methylenetetrahydrofolate production. However, part of the serine is also metabolized via the SERD (serine deaminase) reaction, resulting in the formation of pyruvate and ammonia. This metabolic shift is primarily associated with the increased production of homoserine by *E. coli*, which nearly doubled (from 0.282 to 0.575 mmol·gDCW^−1^·h^−1^) compared to the unmodified model (Appendix A). *M. capsulatus* almost entirely consumes this homoserine via the HSK_GAPFILLING reaction, leading to the formation of O-phospho-L-homoserine. This intermediate is a substrate of the THRS (threonine synthase) reaction for threonine synthesis, which is incorporated into the community. The increased homoserine production results in higher threonine production by *M. capsulatus*. *E. coli* utilizes this threonine in the THRA (threonine aldolase) reaction to synthesize glycine, which is subsequently used in the GHMT2r reaction to regenerate serine (Appendix A). This exchange mechanism provides an alternative pool of pyruvate for *M. capsulatus* and contributes to the observed decrease in acetate production. However, this shift also correlates with the decreased growth rate of the cells. Additionally, sodium ions produced by *E. coli* were identified as another cross-fed metabolite in this community (Appendix A) (a detailed distribution of community model fluxes is provided in the pFBA results available on GitLab at the following link: https://gitlab.sirius-web.org/diploms/2023/esembaeva/-/blob/master/pFBA_data/o2_lim_hom.txt?ref_type=heads (accessed on 18 November 2024).

#### 2.4.2. Nitrate-Limited Conditions

In addition to exploring oxygen-limited conditions, the impact of nitrogen limitation on acetate production in *M. capsulatus* was investigated. Nitrates were selected as the nitrogen source, and a constraint was applied by limiting the flux value of nitrates in the medium to 1.838 mmol·gDCW^−1^. Despite having sufficient oxygen availability, both community models predicted microaerobic conditions for *E. coli* leading to nitrate reduction to nitrites. However, nitrogen exchange between the bacteria differs under these conditions. In the community with the unmodified *i*EC1372_W3110 model, both community members consume nitrates from the medium (Appendix A) (1.406 mmol·gDCW^−1^·h^−1^ by *M. capsulatus* and 0.432 mmol·gDCW^−1^·h^−1^ by *E. coli*) (details: pFBA results available on GitLab at the following link: https://gitlab.sirius-web.org/diploms/2023/esembaeva/-/blob/master/pFBA_data/no3.txt?ref_type=heads, https://gitlab.sirius-web.org/diploms/2023/esembaeva/-/blob/master/pFBA_data/no3_hom.txt?ref_type=heads (accessed on 18 November 2024). In contrast, only *M. capsulatus* consumes nitrates under oxygen-limited conditions. Additionally, *E. coli* produces ammonium under nitrate-limited conditions, while, under oxygen-limited conditions, *E. coli* consumes ammonium. However, only *E. coli* utilizes nitrates as a nitrogen source in the community with the modified *i*EC1372_W3110 model, while *M. capsulatus* uses the nitrites produced by *E. coli*, reducing them to ammonium, which in turn is subsequently consumed by *E. coli* (Appendix A). This interaction is likely linked to amino acid exchanges between the bacteria, which we first compared in order to understand this dynamic better.

Unlike the oxygen-limited conditions, the amino acid composition in this community only consisted of a small amount of homoserine secreted by the *i*EC1372_W3110 model (at a flux of 0.126 mmol·gDCW^−1^·h^−1^) (Appendix A), which was produced regardless of whether the *i*EC1372_W3110 model was modified. This homoserine is almost entirely utilized by *M. capsulatus* in the HSK_GAPFILLING reaction, followed by threonine synthesis via the THRS reaction. However, unlike predictions for oxygen-limited conditions, threonine production is not observed in this scenario (Appendix A). Furthermore, *E. coli* produces a small number of phosphates and sodium ions, which are also consumed by *M. capsulatus* (Appendix A). In addition, some differences in amino acid exchange between the modeling strains in the community with the modified *E. coli* are observed. *E. coli* produces homoserine with a flux of 0.572 mmol·gDCW^−1^·h^−1^, which is almost the same as that observed in oxygen-limited conditions (Appendix A). Homoserine is consumed by *M. capsulatus* via the previously described mechanism, which leads to threonine synthesis. The resulting threonine is incorporated into the community, where it is utilized by *E. coli* in C1 metabolism. As observed in the oxygen-limited community with the modified *E. coli*, the serine produced by *E. coli* is employed by *M. capsulatus* in the GHMT2r reaction and in the SERD reaction for the synthesis of pyruvate and ammonium (Appendix A). The increased production of pyruvate boosts the TCA cycle activity in *M. capsulatus*, leading to the formation of malate and fumarate. The consistent mechanism of amino acid exchange observed under both oxygen- and nitrate-limited conditions, particularly with the active production of homoserine, suggests a significant role of homoserine in modulating *M. capsulatus* metabolism, which influences the metabolic interactions between the bacteria in the community (Figure 3C,D).

The primary carbon source for *E. coli* in the community model with the unmodified *i*EC1372_W3110 is acetate (0.612 mmol·gDCW^−1^·h^−1^) produced by *M. capsulatus*, along with malate (0.126 mmol·gDCW^−1^·h^−1^) and minor amounts of glycerol (6.2 × 10^−^⁴ mmol·gDCW^−1^·h^−1^) absorbed by *E. coli* (Appendix A). The lower availability of these carbon sources compared to that observed in oxygen-limited conditions likely accounts for the differences in the observed growth rates. Meanwhile, *M. capsulatus* also utilizes methane from the medium as its carbon source at a consistent rate. When analyzing the community with the modified *i*EC1372_W3110 model under the same nitrogen-limited conditions, additional differences are noted. Firstly, malate is the primary carbon source for *E. coli*, with its production tripling (from 0.361 to 1.197 mmol·gDCW^−1^·h^−1^) (Appendix A), similar to the pattern seen in oxygen-limited conditions. However, *M. capsulatus* ceases producing acetate and instead produces fumarate, which is absorbed by *E. coli* as an additional carbon source. Malate and fumarate are utilized by *E. coli* in the TCA cycle through the MDH (malate dehydrogenase) and FUM (fumarase) reactions, respectively (Figure 3D). Interestingly, *M. capsulatus* slightly reduces its methane consumption from 18.46 to 17.71 mmol·gDCW^−1^·h^−1^, while the growth rates between the communities under nitrogen-limited conditions remains unchanged. This suggests that the production of pyruvate from serine under these conditions is more effective than that observed under oxygen-limited conditions.

## 3. Discussion

Despite published data supporting acetate secretion by *M. capsulatus* in certain limited conditions, the original *i*McBath model did not demonstrate this. As a result, we analyzed the model’s metabolic pathways and found the absence of acetate transport and exchange reactions. Therefore, we added the corresponding reactions following the example of the *i*IA409 model. Additionally, we set the proportions of fluxes through the RuMP and H_4_MTP pathways according to the study by [8]. A key aspect for the correct functioning of the *i*McBath model is the activity of the Entner–Doudoroff (EDD) pathway, which initially has zero flux. The importance of this pathway is also highlighted in [8], where it is described as one of the key pathways for NADPH and 6-phospho-D-gluconate synthesis, an intermediate product of the EDD pathway. We selected an optimal flux for the reaction based on [8], but additional experimental data are needed in order to confirm the activity of both the EDD and EMP pathways in the strain. Due to the modifications introduced, the *i*McBath model predicted acetate production. However, the rigid constraints on flux bounds artificially narrow the solution space, reducing the model’s flexibility and limiting its applicability in further studies. Therefore, a more detailed investigation of the metabolism of *M. capsulatus* is required to refine and improve the model’s accuracy.

To enable homoserine production while maintaining non-zero biomass in the *i*EC1372_W3110 model, we increased the flux through the pyrophosphate transport reaction (Pitex). OptFlux offered various solutions for homoserine production in *E. coli* (see Appendix A). However, we opted for the Pitex modification because it involved minimal intervention without altering the internal metabolism of the organism.

*M. capsulatus* consumes a greater number of nitrates, which act as electron acceptors during methane oxidation and are reduced to N_2_O under oxygen-limiting conditions. However, the formation of N_2_O does not occur, due to the incomplete description of its reduction process in the *i*McBath model. Thus, it is likely that the *i*McBath model favors the reduction of nitrates to nitrites [53] rather than to ammonium, which explains the excess nitrite release into the medium. These nitrites are partially consumed by *E. coli* along with ammonium, which is produced by *M. capsulatus* in significantly smaller amounts compared to nitrites. Additionally, *E. coli* produces aspartate, which is consumed by *M. capsulatus* as an additional nitrogen source. The *E. coli* strain in the oxygen-limited conditions and homoserine present in the medium consumes nitrates from the environment for homoserine synthesis, reducing its reliance on nitrogen sources produced by *M. capsulatus*. *M. capsulatus* continues to supply ammonia to *E. coli*, as it requires less energy for the cell, while also releasing excess nitrites and ammonia into the medium. Moreover, aspartate ceases to be a metabolite for cross-feeding, and we hypothesize that the reduced growth rate of *M. capsulatus* in this community is linked to the loss of aspartate as an additional nitrogen source.

The opposite situation is observed under nitrate-limited conditions, as follows: *E. coli* becomes the main nitrogen donor, reducing nitrates to ammonia [54] and supplying it to *M. capsulatus* for amino acid synthesis. In the case of modified *E. coli*, the only nitrogen source for *M. capsulatus* in the community is the nitrites produced by *E. coli*, which *M. capsulatus* reduce to ammonia, which is subsequently consumed by *E. coli*. Based on the flux analysis for nitrogen sources in the community, we can also infer that the modified *E. coli* requires a larger amount of nitrogen sources, including nitrates from the medium and ammonium ions, which are utilized for amino acid (serine and homoserine) synthesis. Furthermore, we can assume that this nitrogen exchange is feasible, as both models contain all of the necessary reactions for nitrate reduction to ammonia, while this division of metabolic features helps to reduce the energy costs associated with nitrogen source reduction.

Amino acid exchange is a common occurrence in microbial communities [55,56], and this exchange reduces the metabolic burden on individual strains, helping to maintain optimal growth, particularly under limiting conditions [57,58]. In the study by Wang et al. (2019) [56], interactions within a community of *Dehalobacter restrictus* and *Bacteroides* sp. were analyzed, revealing that *Bacteroides* sp. produce malate when growing on lactate. The malate is consumed by *D. restrictus* for the synthesis of excess NADH. Additionally, *D. restrictus* releases amino acids, such as glutamate, into the medium, which further contributes to NADH production. The generated NADH is utilized by *Bacteroides* sp. for the production of NADPH. The threonine present in the medium is consumed by *D. restrictus* for serine synthesis, illustrating the important role of amino acid cross-feeding. In this microbial community, the production and consumption of malate and amino acids represents a strategy to bypass the impermeability of cell membranes for necessary reducing equivalents (NADH/NADPH) [56]. However, these experiments were conducted with Gram-positive organisms, and further experimental validation is needed in order to confirm similar amino acid exchange in methanotrophic communities involving *E. coli*.

Our hypothesis about serine and threonine exchange in the community is also supported by the fact that *E. coli* uses malate and threonine (reaction THRD with a flux value of 0.434 mmol·gDCW^−1^·h^−1^ and 0.456 mmol·gDCW^−1^·h^−1^ under nitrogen- and oxygen-limited conditions, respectively) provided by *M. capsulatus* for NADH synthesis. The serine produced by *E. coli* is consumed by *M. capsulatus* for pyruvate synthesis via serine-pyruvate aminotransferase activity [59], which is employed to produce the malate necessary for *E. coli*. Serine, in turn, is an important metabolite for *M. capsulatus*, due to its use in the partial serine cycle. Additionally, mutants lacking serine-glyoxylate aminotransferase (SGAT), which in *M. capsulatus* also possesses serine-pyruvate aminotransferase activity, did not show reduced growth rates, but they did exhibit a longer lag phase [59,60].

Thus, we hypothesize that it is more advantageous for the community to utilize malate instead of acetate in the microbial community with modified *E. coli*, due to its increased NADH demand. Since the acetate negatively affects the growth rate of *M. capsulatus* [1], such an interaction in the community may be beneficial by reducing the acetate levels in the medium.

The community-level metabolic modeling approach used in this study is useful to describe how the microbial consortium functions in different growth conditions, to model optimal medium composition for efficient growth, and to identify relationships and functionality within the community. However, this approach is a non-optimizing modeling method used to design natural and synthetic microbial communities, not considering the spatial organization of the community or the integration of different omics data (transcriptomics, proteomics, metabolomics, and fluxomics) with CBMc models, which is required to constrain the models in order to precisely simulate wider scenarios of consortium functioning.

## 4. Materials and Methods

### 4.1. GSM Model for Methylococcus capsulatus

To reconstruct the community model, we used a published *i*McBath model [3] for *Methylococcus capsulatus*. However, the model required a list of modifications, which were performed via the CobraPy library (v. 0.25.0; [50]). The program code presented is in Jupyter Notebook and available at a common link (https://uni.sirius-web.org:58443/bioumlweb/#de=data/Collaboration/sysbio2024_Esembaeva/Data/A_study_of_the_community/iMcBath_model.ipynb (accessed on 18 November 2024)). Two isoforms of methane monooxygenase, sMMO and pMMO, are encoded in the *M. capsulatus* genome. The functionally active MMO in the presence of Cu^2+^ in the medium is pMMO. To consider it, we took off the sMMO reactions from the community model. We selected the uphill electron transport mechanism for the *i*McBath model as the most likely one, based on studies in which the pMMO was demonstrated to be the main methane monooxygenase with a proposed electron transfer mode [3,44]. A summary table with the necessary reactions and their modifications is presented in Appendix A. To compare the flux balance analysis results with the original study, a pFBA method [61] implemented in the CobraPy library was used. The flux balance analysis across different electron transport scenarios is provided in Appendix A.

We observed that the *i*McBath model lacked acetate transport and exchange reactions. To correct this, we extended the model by ACK2r (acetate transport) and EX_ac_e (acetate exchange) reactions, setting the reaction boundaries [0, 1000] to allow for acetate secretion, following the example of the reactions in the *Methylotuvimicrobium alcaliphilum* 20Z^R^ model (*i*IA409) [62]. To access the production of the acetate under oxygen- and nitrate-limited conditions, the model needed a number of extra modifications. Firstly, we analyzed the key metabolic pathways in the *i*McBath model and compared it with published models for both *M. alcaliphilum* 20Z^R^ (*i*IA409) and *M. capsulatus* (*i*MC535) [8]. Based on similar reactions in both *i*IA409 and *i*MC535 models, we adjusted the directions of reactions in the TCA cycle, EMP, ribulose monophosphate pathway (RuMP), and Entner–Doudoroff (EDD) pathway in the *i*McBath model. The ratio of RuMP to H4MTP pathways was set to 1.2, and the flux through the EDD pathway was fixed at 6 mmol·gDCW^−1^·h^−1^, according to previous publications [8,63,64]. A list of reactions and their modifications is provided in Appendix A. A flux balance analysis of the modified model was also performed using the pFBA module in the CobraPy library. The resulting fluxes for different values of oxygen and nitrogen availability are presented in Appendix A. The model and all pFBA results are also available via GitHub at the following address: https://gitlab.sirius-web.org/diploms/2023/esembaeva (accessed on 18 November 2024).

To add the capability for homoserine uptake in the community model, we modified the homoserine transport and exchange reactions based on the example of the original *E. coli* model (*i*EC1372_W3110) [65], namely the EX_hom__L_e, HOMtex, and HOMt2pp reactions with the following corresponding boundaries: [0, 1000]; [−1000, 1000]; and [−1000, 1000].

### 4.2. GSM Modeling of Homoserine Production in E. coli

As a satellite bacterium for the community model, an *E. coli* strain with a corresponding GSM model (*i*EC1372_W3110) was used. The model demonstrated the potential for homoserine production, as shown by Vo and co-authors [18], and experimentally confirmed by the *E. coli* W3110 strain’s growth. To achieve homoserine production and excretion, we modified the following two reactions: homoserine transport (HOMt_atp) and phosphate acetyltransferase (PTAr2). The homoserine transport reaction was adjusted to require ATP, and an ATP molecule was added to the PTAr2 reaction for acetyl-CoA synthesis, according to [18]. The previously published results on homoserine production [18] were reproduced via setting the production reaction as the objective function and combining the activity of aspartate-forming reactions (L-aspartase (ASPT) and aspartate transaminase (ASPTA) with acetate-consuming reactions (acetate kinase (ACKr) and acetyl-CoA synthetase (ACS)).

To optimize the model solution, the CBMc model’s variant with the highest homoserine secretion (10.22 mmol·gDCW·h^−1^) was selected, with active L-aspartase and acetyl-CoA synthetase pathways. To identify the necessary modifications in the model for homoserine secretion and non-zero biomass, the OptFlux (v. 3.4.0; [47]) optimization tool was used, with 2500 evolutionary generations, the SPEA2 algorithm, and the WYIELD objective function. The pFBA method was employed to reproduce the optimization in the model. The solutions proposed by OptFlux were incorporated into the model using the CobraPy library.

For the unmodified *E. coli* model, we used a version with modified HOMt_atp and PTAr2 reactions, but without an alteration of the flux through the phosphate transport reaction.

### 4.3. CBM Modeling of the iMcBath and iEC1372_W3110 Community

To combine the *i*McBath and *i*EC1372_W3110 models into a community model, we used the PyCoMo tool (version: 0.2.2; [31]). We used a modified version of the *M. capsulatus* model with the acetate production described above. As a model for non-homoserine conditions, the *i*EC1372_W3110 model with only modified reactions for homoserine transport was employed. To consider the homoserine production conditions, the *i*EC1372_W3110 model with simultaneous production of homoserine and growth rate was harnessed. The growth environment for the community was designed by creating a dictionary of essential medium components required for both *i*McBath and *i*EC1372_W3110 models, with methane as the sole carbon source. Using the ‘apply_fixed_abundance’ method, we set the *M. capsulatus* to *E. coli* ratio to 9:1, based on the ratio found in the article [15]. To establish limiting conditions, we reduced the amount of oxygen or nitrates in the medium composition dictionary for the community model. Additionally, the consumption reactions for co-lipids, biotin, glycolate, and O-phospho-L-serine in the community model had to be restricted, since these compounds were neither included in the medium nor produced by any model in the community. Proton exchange reactions were also limited, with boundaries set to [−1000, 0] for the uptake from the medium. The CO_2_ exchange reaction in the *E. coli* model was restricted to CO_2_ release [0, 1000]. Furthermore, the strict boundaries were set for methane uptake by the *i*McBath model [−18.46; −18.46] under oxygen-limited conditions. All boundaries were established using the ‘bounds’ method. All reactions and their boundaries are detailed in Appendix A. A reconstructed workflow is available on the BioUML platform dedicated to modeling biological systems [43] (link to the Jupyter Notebook: https://uni.sirius-web.org:58443/bioumlweb/#de=data/Collaboration/sysbio2024_Esembaeva/Data/A_study_of_the_community/gui_reconstruct_community_model.ipynb (accessed on 18 November 2024)).

#### Application of the Developed GUI for Community Model Reconstruction


**Step 1. Upload models**


The first step involves the selection of a number of metabolic models that will comprise the community model. Depending on the chosen number, a corresponding number of selectors will appear. For ease of use, by clicking the “Upload” button, it is possible to upload a model in XML format or select the models used in this research that are presented in the selector. It is also necessary to enter names for the selected models.

After completing these steps, the “Load models” button should be clicked to save the models. Since the investigation focused on the effect of homoserine on the *M. capsulatus* and *E. coli* community, a checkbox was added to include the transport and exchange reactions of the homoserine in the *M. capsulatus* model. This checkbox should be applied after loading the models. The interface of the steps is presented in Appendix A.


**Step 2. Reconstruct a community model**


The second step represents the reconstruction of the community model based on the user-uploaded models. To do this, it is necessary to click the ‘Create community’ button, which is shown in Appendix A, and anticipate the model reconstruction process to complete it. It can take up to 30 min to create a community model consisting of two bacteria.


**Step 3. Set up microbial abundance and the community growth medium**


The next step allows for adjusting the abundance of microorganisms in the community model using sliders. The total proportion of microorganisms must sum to 1, which should be considered during the setup. This specification should be confirmed by pushing the “Apply abundance” button (Appendix A).

The environment configuration involves selecting a reaction for modification through a reaction selector. The bounds are set, followed by applying the changes via the “Apply fluxes” button. In cases where a reaction needs to be removed, it has to be selected from the selector, followed by the “Remove reaction” option. After modifications are performed, the “Apply medium” button is pressed to apply the medium in the community model (Appendix A).


**Step 4. Modify required reactions in the community model**


To modify reaction bounds in the community model, a reaction search and selector have been added. After selecting a reaction, its upper and lower bounds can be set, followed by pressing “Apply bounds”. If multiple reactions need to be modified, the process can be repeated. Once all changes are complete, they must be confirmed by pressing the “Confirm changes” button. Pressing the “Model summary” button will display the model calculation (Appendix A).


**Step 5. Download the resulting community model**


The final step is saving the obtained community model and exporting it to SBML format, as well as downloading the pFBA results for the community model in TSV format. The results are downloaded via a link through Jupyter Notebook (Appendix A).

## 5. Conclusions

In this study, we developed a pipeline with the GUI interface for the reconstruction of community metabolic models and applied it to develop several community models under nitrogen- and oxygen-limited conditions for *Methylococcus capsulatus*, a model methanotroph used in biotechnology as a single-cell protein producer. *Escherichia coli* W3110 was harnessed as a potential satellite microorganism for *M. capsulatus*. We also built community models with *E. coli* W3110 modified for homoserine production to analyze its impact on the community. Our study has demonstrated that the presence of homoserine in the community leads to a reduction in acetate secretion by *M. capsulatus*. This, in turn, may enhance resource efficiency and increase protein yield during the industrial cultivation of *M. capsulatus*. The pipeline that we developed for constructing community metabolic models provides deeper insights into microbial interactions under various conditions. It could serve as a foundation for further research and optimization of microbial communities like multicomponent lactic acid bacterium strains and yeasts aimed to increase their productivity or fermentation characteristics. However, the modeling still requires experimental validation to confirm the exchange rates for cross-feeding metabolites and to verify the predicted growth rates of the community members.

## Figures and Tables

**Figure 1 ijms-25-12469-f001:**
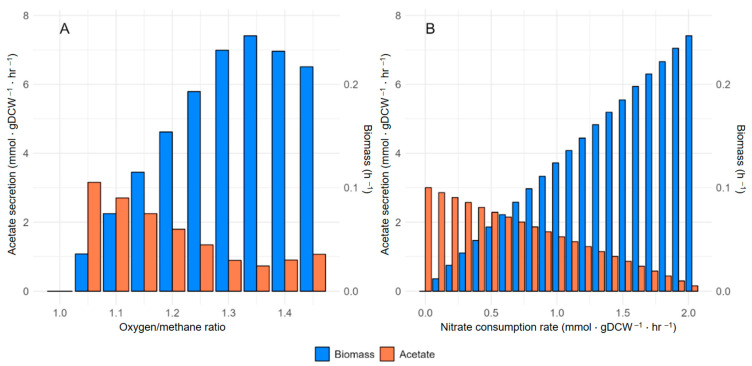
(**A**) The impact of the oxygen/methane ratio on acetate production predicted by the *i*McBath model after modifications. (**B**) The effect of nitrate reduction in the medium on acetate production predicted by the *i*McBath model following the modifications.

**Figure 2 ijms-25-12469-f002:**
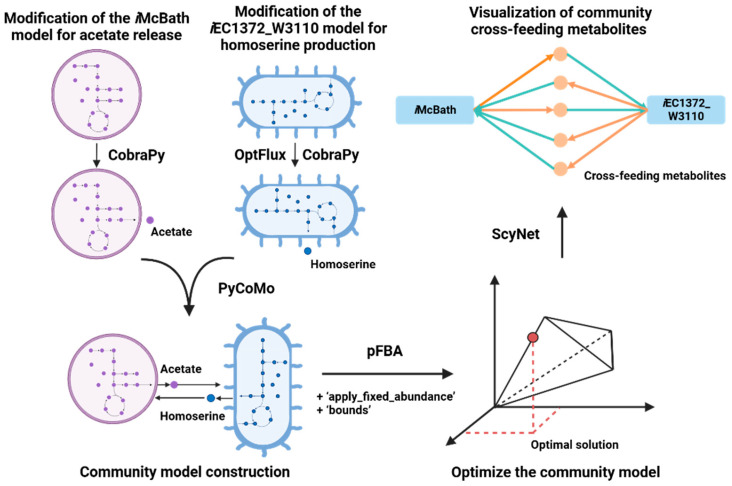
Schematic representation of the developed pipeline in the BioUML platform [43] for the reconstruction and analysis of the *M. capsulatus* and *E. coli* microbial community models. Initially, the original *i*McBath and *i*EC1372_W3110 models were modified using the COBRApy library [50] and the OptFlux tool [47] to enable acetate secretion in *i*McBath and homoserine production in the *i*EC1372_W3110 model (see Section 4). These models are combined into a community model using the PyCoMo tool [31]. Subsequently, modifications to the community model can be performed using the ‘apply_fixed_abundance’ and ‘bounds’ methods with followed up parsimonious flux balance analysis (pFBA) in COBRApy (see Section 4). Finally, fluxes for cross-feeding metabolites within the community are visualized using ScyNet [52], where the color shows which metabolites are consumed (turquoise lines) and which are excreted (orange lines) by community members from (and to) the environment.

**Figure 3 ijms-25-12469-f003:**
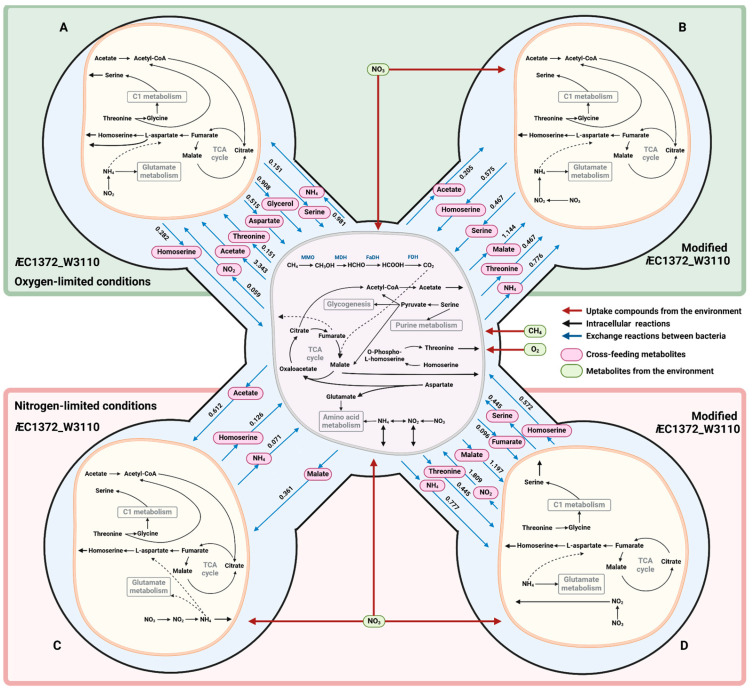
Schematic representation of cross-feeding metabolites in the community consisting of *M. capsulatus* and *E. coli* under oxygen-limited conditions (**A**), *M. capsulatus* and modified *E. coli* under oxygen-limited conditions (**B**), *M. capsulatus* and *E. coli* under nitrogen-limited conditions (**C**), and *M. capsulatus* and modified *E. coli* under nitrogen-limited conditions (**D**). *M. capsulatus* is shown in purple, while *E. coli* is shown in yellow. Oxygen-limited conditions are highlighted with a green background, whereas nitrogen-limited conditions are represented with a pink background. Metabolites from the medium are indicated in green, and cross-feeding metabolites are shown in pink. Red arrows represent metabolite uptake from the medium, black arrows indicate internal reactions of the models, and blue arrows indicate exchange reactions. The numbers above the blue arrows indicate metabolite flux with units of mmol·gDCW^−1^·h^−1^, with flux values taken from Appendix A.

**Table 1 ijms-25-12469-t001:** Modifications of the *i*EC1372_W3110 model for homoserine production.

Model ID	Modifications	Growth Rate, h^−1^	Homoserine Production, mmol·gDCW^−1^·h^−1^
W3110	ACKr, ASPTA	-	8.47
W3110	ACKr, ASPT	-	8.70
W3110	ACS, ASPTA	-	9.97
W3110	ACS, ASPT	-	10.22
W3110_OptFlux	ACS, ASPT, Pitex	0.37	5.37

**Table 2 ijms-25-12469-t002:** Predicted growth rates of the community, as well as growth rates of *E. coli* and *M. capsulatus* strains in the in silico community.

Community Scenario	Community Growth Rate, h^−1^	*M. capsulatus* Growth Rate, h^−1^	*E. coli* Growth Rate h^−1^
Oxygen-limited, unmodified *E. coli*	0.225	0.203	0.022
Oxygen-limited, modified *E. coli*	0.186	0.167	0.019
Nitrate-limited, unmodified *E. coli*	0.217	0.195	0.022
Nitrate-limited, modified *E. coli*	0.217	0.195	0.022

## Data Availability

Jupyter Notebooks, models, and datasets with results are available on GitLab via the following link: https://gitlab.sirius-web.org/diploms/2023/esembaeva (accessed on 18 November 2024), as well as on BioUML via the following link: https://uni.sirius-web.org:58443/bioumlweb/#de=data/Collaboration/sysbio2024_Esembaeva/Data/A_study_of_the_community (accessed on 18 November 2024), where the GUI is also accessible.

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
