# Peer review of "A Study of the Community Relationships Between Methanotrophs and Their Satellites Using Constraint-Based Modeling Approach"

_ijms, 2024, doi:10.3390/ijms252212469_

Round 1

Reviewer 1 Report

Comments and Suggestions for Authors

11)      The ending of Abstract section has to somehow be more specific and deterministic on authors’ own study, analysis, methods and findings, since the current text content sounds rather “general and introductory”, not case specific.

22)      Either the existing text length of section 1. Introduction can be separated into two shorter subsections in which the densely presented theoretical coverage to be conveyed in a more spacious and comprehensive manner, or authors can develop another main section 2.Literature Review, thus, concisely representing the roles and the contribution of: methanotroph microorganisms to biotechnology  and that of heterotrophic bacteria at  the genome-scale metabolic models (GSMs).

33)      Considering the research objectives and the scopes of authors’ study:

CBMc models reconstruction to analyze the interactions between Methylococcus capsulatus and an engineered E. coli strain under oxygen and nitrogen-limited conditions in bioreactors. Our objective is to evaluate the potential of E. coli to reduce methanotroph by-products, particularly acetate, and assess its impact on the growth and metabolism of Methylococcus capsulatus using a community metabolic modeling approach”,

we developed a pipeline with GUI interface for reconstruction of community metabolic models and applied it to develop several community models under nitrogen- and oxygen-limited conditions for Methylococcus capsulatus, a model metha-566 notroph used in biotechnology as a single cell protein producer”,

I am skeptical about the close relevancy, or not, of the study to the themes and topics of Int. J. Mol. Sci. The “molecular” dimension of the study can be better developed in order to be better matched with the journal’s thematics.

44)      The section of Materials and Methods has to be placed after the missing 2.Literature Review section, not as section 4.Besides, the input data and the quantitative information (based on the 3 subsections 4.1,4.2,4.3) cannot be purely descriptive in plain text, but it can be better to be collectively presented in the form of Table(s).

55)      Besides to the subsections devoted to Oxygen-limited and Nitrate-limited conditions authors are recommended to succinctly discuss what are the applications and the importance of their findings of the specific CBM modeling beyond the specifically applied conditions, towards a wider and real world conditions applications? Could be scalability constraints or other barriers from generalized the findings to other applications of similar interest? Are there synergistic effects with similar technologies?

Author Response

First of all, we would like to thank the reviewers for their critical comments and helpful suggestions.

Based on these comments and suggestions, we have made careful modifications to the original manuscript. The reviewer’s comments are shown in black, followed by our responses in blue. The modifications made to the manuscript following the comments are marked by yellow in the revised manuscript and Supplementary materials.

Reviewer 2 Report

Comments and Suggestions for Authors

In this study, the authors developed community metabolic models of the methanotroph M. capsulatus and E. coli and simulated their growth and interactions under different nutrient-limiting conditions. I have a few questions and comments regarding this work.

Major Comments

1. In the Abstract and Introduction, there is no need to mention or comment on other tools for community metabolic modeling, as this work does not introduce a new tool.

2. In oxygen-limiting conditions, is the value 22.152 referring to oxygen concentration (L176) or oxygen uptake rate (L208)? Can concentration and flux have the same value? A similar question applies to nitrogen content/flux.

3. What does the ScyNet visualization (L199) depict?

4. The statement in L222-224 is not reflected in the mentioned figures.

5. I suggest adding an integrated flux map in the Results section to compare the four community scenarios.

6. Could the model simulate acetate inhibition in M. capsulatus? Would reducing acetate levels in the media benefit growth?

Minor Comments

1. The values in Table 1 should display a consistent number of decimal places. Does “-” indicate zero growth? The first and last models appear to be duplicates.

2. The code provided on GitLab is not accessible.

Comments on the Quality of English Language

The quality of English language is acceptable.

Author Response

(The authors gave the same response as above.)

Round 2

Reviewer 1 Report

Comments and Suggestions for Authors

At this revised manuscript authors considered the review comments and addressed them in a satisfactory manner. The study has been well organized, providing new novel findings in the fields of biotechnology and methanotrophy of methane-utilizing bacteria with the potential for enhancing microbial consortia productivity. The study focused on those microbial consortium functions in different growth conditions that can model optimal medium composition. In this perspective, the research outcomes can guide for target genetic modifications and optimization of metabolic pathways for various industrial applications. In this context the revised manuscript can be accepted for publication at the International Journal of Molecular Sciences as is.

Reviewer 2 Report

Comments and Suggestions for Authors

The authors have addressed my questions. I have no further questions.